# Preparation and Characterizations of PSS/PDADMAC Polyelectrolyte Complex Hydrogel

**DOI:** 10.3390/polym14091699

**Published:** 2022-04-21

**Authors:** Thichakorn Sungoradee, Kawee Srikulkit

**Affiliations:** 1Petrochemistry and Polymer Science, Faculty of Science, Chulalongkorn University, Bangkok 10330, Thailand; 6072866723@student.chula.ac.th; 2Department of Materials Science, Faculty of Science, Chulalongkorn University, Bangkok 10330, Thailand; 3Center of Excellence on Petrochemical and Materials Technology, Chulalongkorn University, Bangkok 10330, Thailand

**Keywords:** polyelectrolyte complex hydrogel, porosity, viscoelasticity, absorbency, textile dye removal

## Abstract

Polyelectrolyte complex (PEC) hydrogel, formed via physically electrostatic crosslinks between polyanion and polycation, is an interesting hydrogel in terms of its nontoxicity and solvent-free technique. In this work, poly (sodium 4-styrenesulfonate) (PSS)/poly (diallyl dimethyl ammonium chloride) (PDADMAC) complex hydrogels were prepared. Firstly, the PSS/PDADMAC complex aggregates using various PSS/PDADMAC mole fractions that were prepared in the presence of NaCl solution. Then, the aggregates were resolubilized under stirring at 70 °C for 2 h to obtain a homogeneous PEC solution. Finally, the PEC solution was dialyzed using a dialysis membrane with 3500 molecular cut-off for 1 day. The dialysis bath was changed every interval period of 2 h to control the rate of reversible electrostatic interaction, resulting in the homogenous PEC hydrogel with porous morphology as revealed by SEM and BET investigations. The dimensional stability and viscoelasticity of the PEC hydrogel was studied by DMA experiment, which showed the viscoelastic behavior at a compressive force ranging from 0 to 0.1 N. Finally, PSS/PDADMAC hydrogels showed a high water absorbency property and excellent affinity to textile anionic dyes.

## 1. Introduction

Hydrogels are three-dimensional network polymers, which are achieved by either physical or chemical crosslinks [1]. Hydrogels exhibit interesting properties such as high-water uptake, separation, scaffolds, control release, biomedical properties, electroresponsive properties, as well as having an electrochemical performance. Therefore, they could offer potential applications in various fields, including environmental remediation [2,3,4,5,6,7], membrane [8,9,10], tissue engineering [11,12], drug release and drug delivery [13,14,15,16,17,18,19,20,21,22], sensors [23,24,25,26,27,28,29], and supercapacitors [30,31,32,33,34]. Polyelectrolyte complex hydrogels (PEC hydrogels) belong to physical crosslinks such as hydrogen bonding, van der Waals, and ionic (electrostatic) interactions. Examples of PEC hydrogels include biobased hydrogels such as polysaccharide hydrogels, which can be prepared from cationic/anionic biopolymers, such as cationic/anionic polysaccharide complexes [35,36,37]. Other PEC hydrogels are synthetic polyelectrolyte complexes between an anionic polyelectrolyte such as poly (sodium-4-styrene sulfonate) (PSS) and a cationic polyelectrolyte such as poly (dimethyl diallyl ammonium chloride) (PDADMAC) [38]. Advantageously, PECs are the complexes formed between oppositely charged polymers combined with H-bonding, van der Waals forces, and dipole interactions, leading to the formation of networks without the chemical cross-linkers, thereby reducing the possible toxicity and other undesirable effects of reactive reagents.

The process of the formation of PEC hydrogels can be divided into three steps. Firstly, the primary complex aggregate (i) forms immediately in an uncontrolled manner once oppositely charged polyelectrolyte solutions are mixed, leading to PEC aggregate. Then, the complex aggregate can be resolubilized again in the presence of electrolytes such as NaCl, resulting in a viscous PEC solution (ii). The electrolyte plays a role in swelling the PEC aggregate by shielding the anionic polyelectrolyte against the cationic polyelectrolyte to prevent the reformation of the aggregate. At the critical content of the electrolyte, the PEC solution is obtained. Lastly, the PEC hydrogel (iii) is achieved by controlling the dilution of the electrolyte concentration via the dialysis process. Gradual diffusion of the electrolyte slowly causes phase separation, arising from the reformation of electrostatic interaction. Finally, the PEC hydrogel is obtained [38]. There are many factors affecting the structure, dimensional stability, and the mechanical and physical properties of PEC hydrogels, such as polyelectrolyte mole fractions, polyelectrolyte molecular weights, polyelectrolyte polymer structure, temperature conditions, dialysis membrane, and ionic strength.

In summary, the key factors affecting the PEC hydrogel structure are the polyanion/polycation mole ratio, salt concentration, and desalting process. The PEC aggregate in the presence of salt causes the aggregate to be swollen; the greater the salt concentration, the larger the size of the swollen aggregate, consequently resulting in the homogenous PEC solution. In the presence of salt, it can be explained that salt plays a role in dispersing the ordered PE/PE complexation into the coiled structure. The formation of the PEC hydrogel is then obtained by gradual desalting through the dialysis technique, which causes phase separation, creating a porous structure with a cell wall composed of the PE–PE complex. So far, little information on PEC formation, structure, porosity, and compressive strength has been reported.

Therefore, the scope of this work was to prepare a PSS/PDADMAC-based PEC hydrogel. The effects of the stoichiometric fractions and the desalting process on the hydrogel properties (dimensional stability, water uptake, morphology, strength, and porosity) were evaluated.

## 2. Materials and Methods

### 2.1. Materials

Poly (sodium-4-styrene sulfonate) (PSS, Mw 70,000 g/mol) and 20% (*w/w*) poly (dimethyl diallyl ammonium chloride) (PDADMAC, Mw 200,000–350,000 g/mol) (Figure 1) were purchased from Sigma Aldrich. Sodium chloride (NaCl) and dialysis membrane with molecular cut-off of 3500 were obtained from QRëC, and Spectrumlabs, respectively.

### 2.2. Preparation of PEC Hydrogel

PSS/PDADMAC PEC hydrogels having various PSS:PDADMAC mole fractions were fabricated. In this experiment, 7 series of PSS/PDADMAC PEC aggregates were prepared as follows: PSS0.80:PDADMAC0.20, PSS0.70:PDADMAC0.30, PSS0.60:PDADMAC0.40, PSS0.50:PDADMAC0.50, PSS0.40:PDADMAC0.60, PSS0.30:PDADMAC0.07, and PSS0.20:PDADMAC0.80, which were designated to f0.8, f0.7, f0.6, f0.5, f0.4, f0.3, and f0.2, respectively. The compositions of chemicals used for PSS/PDADMAC hydrogel preparation are summarized in Table 1. First, PSS/PDADMAC swollen aggregates were synthesized by mixing PSS/NaCl solution with 20 wt% PDADMAC solution. Second, the as-prepared swollen aggregates were resolubilized by continuous stirring at the temperature of 70 °C. Finally, the PSS/PDADMAC solutions were dialyzed using dialysis membrane tube to obtain PSS/PDADMAC PEC hydrogels. An example of PSS0.30:PDADMAC0.70 hydrogel (f0.3) was prepared as follows: 50 mL of 0.3 mole PSS containing 3 M of NaCl solution were prepared in a 250 mL beaker, and then 28.3 g (0.7 mole) of 20 wt% PDADMAC solution was slowly added, resulting in the formation of swollen aggregate, as shown in Figure 1 [39]. Then, the as-prepared aggregate was resolubilized under stirring at 70 °C for 2 h. Note that the solution volume was adjusted to 50 mL prior to dialysis. Then, dialysis was carried out by desalting method using dialysis membrane tube in a deionized water bath for 24 h (interval period of 2 h for bath change) to obtain the PSS/PDADMAC PEC hydrogel.

### 2.3. Characterization Techniques

Scanning electron microscopy (SEM) was performed using scanning electron microscope (SEM JEOL model JSM-6400LV, Japan) at an accelerating voltage of 15 kV. Samples were freeze-dried using Christ model beta 1–8 LD plus and kept dried prior to characterization. Dynamic mechanical analysis (DMA) was carried out using DMA861e Mettler Toledo. Hydrogel with diameter of 8 mm was prepared. Characterization conditions were as follows: testing mode: compression mode, displacement amplitude: 316 µm, and testing temperature: 25 °C. Surface area is determined using Micromeritics BET analyzer (model 3Flex). All freeze-dried samples were cleaned by N_2_ gas desorption at temperatures between 150 °C and 200 °C prior to testing. N_2_ gas adsorption at the temperature of liquid state was performed to obtain adsorption isotherm. Water absorption was carried out by immersion method using a freeze-dried sample. The triplicate samples before immersion (dry) and after immersion were weighed and subsequently calculated and averaged as follows: percent water uptake (%) = (W_af_ − W_bf_)/W_bf_ × 100, where; W_af_ = hydrogel weight; W_bf_ = dry weight.

C.I. Reactive Blue 49 (an anionic dye for dyeing cellulose) adsorption onto freeze-dried PEC hydrogels was carried out. First, 50 mL of dye solutions (50 ppm, 100 ppm, and 150 ppm) was prepared in a 250 mL Erlenmeyer flask for each hydrogel sample. Then, 25 mg of freeze-dried PEC hydrogel was added into the prepared flask. The flask was agitated (85 rpm at room temperature) on a shaker to investigate adsorption kinetic. A total of 3 mL of dye solution was taken at 60 min. interval time to measure residual dye absorbance using LabTech (BlueStar B) spectrophotometer at λmax = 590 nm. Note that after measuring residual dye absorbance, the sample was put back into the Erlenmeyer flask again. Then, percent dye adsorption (% dye adsorption) vs. contact time (t) was measured. The adsorption capacity Q_t_ (mg/g) at a certain contact time (t) was calculated and plotted.

## 3. Results and Discussion

### 3.1. Mechanism of PSS/PDADMAC PEC Hydrogel Formation

Typically, the PSS/PDADMAC PEC complex is present in an aggregate form. However, the PSS/PDADMAC aggregate can be converted into PSS/PDADMAC hydrogel, as described earlier. The structure and properties of PSS/PDADMAC hydrogel are greatly dependent on the stoichiometric mole ratio between polycation and polyanion. In the case of polyelectrolytes, their molecular weights are defined by the average molecular weight (Mw or Mn), which cannot be used for the calculation of the stoichiometric mole ratio. Therefore, in this experiment, the stoichiometric mole ratio of ions (cation and anion) was calculated based on the mole of the monomers (sodium styrene sulfonate (SS, Mw = 206 g/mole) and diallyl dimethyl ammonium chloride (DADMAC, Mw = 161 g/mole)). The mechanism of the PSS/PDADMAC PEC hydrogel formation can be divided into three steps (aggregate formation, solubilization of the aggregate, and PEC hydrogel formation). At the first step, polycation and polyanion polyelectrolytes are mixed. As a result, phase separation due to the polyelectrolyte complex formation immediately occurs, resulting in an aggregate, as seen in Figure 1. At the solubilization step (heating/stirring), the NaCl concentration and temperature play a role in dispersing the electrostatic interaction; the greater the salt concentration, the greater the dispersibility. At this step, Na^+^ and Cl^−^ in the aqueous phase are re-absorbed into the aggregate. The absorption rate is enhanced by mechanical and thermal treatment, as illustrated in Figure 2. The minimum concentration of NaCl solution that completely dissolved PSS/PDADMAC aggregate was carried out. In the case of f0.3 (referred to Figure 1), the solubility test was investigated in the presence of 1 M, 2 M, and 3 M NaCl. It was found that the PSS/PDADMAC aggregate was only swollen in the presence of 1 M NaCl, partially soluble in the presence of 2 M NaCl, and completely soluble in the presence of 3 M NaCl at the fixed temperature of 70 °C. Therefore, 3 M NaCl was chosen in this case. From Table 1, the minimum NaCl concentrations for f0.2-f0.4 and f0.5-f0.8 are 3 M and 4 M, respectively. It was found that f0.4 and below require a lower minimum NaCl concentration than f0.5 and above. These results imply that the electrostatic strengths of f0.5 and above are higher than those of f0.4 and below, hence requiring the higher minimum NaCl concentration to separate the polycation and polyanion. At the desalting step, the PEC hydrogel was slowly formed, due to the reversible formation of the electrostatic force between the polycation and polyanion, resulting in phase separation. After the desalting process was complete, the viscoelastic hydrogels are obtained, as shown in Figure 3.

### 3.2. Physical Appearance of PSS/PDADMAC PEC Hydrogels

The obtained PSS/PDADMAC PEC hydrogel representatives are illustrated, as shown in Figure 3. As can be seen, all of the hydrogels are opaque except for f0.2, which is translucent and was never dried, due to its hygroscopicity. PDADMAC is called polysalts, which is a hygroscopic polymer by its nature. Therefore, hydrogels containing high ratios of PDADMAC exhibit not only hygroscopicity but also dimensional instability due to the mobility of the hydrogel cell wall. On the other hand, PSS, which contains a rigid aromatic pendant group, is responsible for the dimensional stability of PSS/PDADMAC hydrogels with high PSS mole fractions, as observed in Figure 3. Therefore, the f0.2 hydrogel is not suitable for the applications where dimensional stability is concerned. The opacity of the hydrogels is derived from their porous structure, which was confirmed by the SEM images. The volume size and shape are found to be dependent on the mole fraction. For the f0.5 representative, the hydrogel exhibits compactness and stiffness of structure, arising from the strongest electrostatic interaction at the charge balance point [38]. At PSS mole fractions above f0.5 (f0.6-f0.8), the net charge balance is negative. An increase in PSS mole fraction (f0.6-f0.8) results in an increase in volume and size, with excellent dimensional stability arising from the repulsive interaction among PSS sulfonate anions. On the other hand, at PSS mole fractions below f0.5 (f0.4-f0.2), the net charge balance is positive, resulting in a hygroscopic PE–PE complex cell wall (discussed later), which leads to hydrogels with poor dimensional stability, particularly f0.3 and f0.2. Moreover, due to the hygroscopic nature of the PDADMAC, the hydrogel sample (f0.2) which had the highest content of PDADMAC was present in a never-dry gel form.

### 3.3. Scanning Electron Microscopy

During the desalting step using the dialysis membrane, the reversible electrostatic interaction slowly formed, causing a gradual phase separation to produce the cell wall of the PEC hydrogel, as shown in the SEM images (Figure 4). At the initial desalting step, the phase separation quickly occurs at the surface of the dialysis membrane tube, forming a solid-like PEC hydrogel at the inner side of the dialysis tube, which helps control the diffusion rate of the electrolyte. As the solid-like PEC hydrogel was obtained, the dialysis was stopped, and the PEC viscoelastic hydrogel was removed. To observe the porous structure, the sublimation of the iced water crystals, which resembled pores due to the freeze-drying process, was carried out prior to SEM analysis. As can be seen from the SEM images of the freeze-dried samples (Figure 4), the porous structure comprising macropores (arising from phase separation) and micropores (arising from water re-absorption into the cell wall) is observed. In the case of the f0.8 and f0.7 samples, macropores with a thick and dense cell wall are observed. The thick and dense cell wall is derived from the hydrophobic-rich aromatic pendant groups of PSS, which deter the water re-absorption into the cell wall. Therefore, in these cases, the micropores on the cell wall are not observed. In comparison, the f0.6 and f0.5 samples exhibit much smaller and denser macropores, which exhibit a porous, thin cell wall (micropores), indicating that the cell wall formation as well as the water re-absorption into the cell wall gradually occur. A further decrease in PSS content (f0.4-f0.2) results in a porous, thin cell wall with large macropores. As can be seen, the micropores are large due to the hygroscopicity of the cell wall containing dominant PDADMAC ratios, which allow for more water re-absorption into the thin cell wall than in the case of the f0.8-f0.5. As presented in Figure 2, hydrogels with a PSS content below f0.5 exhibit dimensional instability as well as poor mechanical properties, due to their large macropores combined with their porous, thin cell wall (large micropore). The macropore size of the (f0.8-f0.2) hydrogels is summarized in Table 2.

### 3.4. Surface Area Determination by BET Analysis

The surface area is one of the most important properties of porous materials which are widely investigated in various fields of applications, such as pollution remediation, catalysts, separation materials, drug delivery, tissue engineering, and sensors. In this study, the surface area determination of the freeze-dried PEC hydrogels was carried out using a Micromeritics BET analyzer. The BET analysis was performed based on the adsorption isotherms of nitrogen gas molecules at a range of pressures that caused a formation of the monolayer coverage of N_2_ molecules. The obtained isotherms are transformed into the linear BET plot (in a region of P/Po range from 0 to 0.3). Thereafter, the surface area was determined and summarized in Table 3. The results show that the BET surface area decreases with a decrease in the PSS mole fraction. The surface areas of f0.8, f0.7, f0.6, f0.5, f0.4, and f0.3 samples are 3.54, 2.16, 1.72, 0.98, 1.19, and 0.48 m^2^/g, respectively. Note that, at the stage of the sample preparation using N_2_ gas to clean the surface at a high temperature under vacuum, the delicate sample, f0.3, which has large macropores with porous, thin cell wall, was prone to collapse, resulting in a significant decrease in the surface area determined by BET analysis. Therefore, the BET surface areas of delicate hydrogels, particularly f0.4 and f0.3, might not represent the true values. Note that the BET surface area of f0.2 was not measurable for that reason.

### 3.5. Compressive Strength by Dynamic Mechanical Analysis

Apart from dimensional stability, the dynamic properties of the viscoelastic materials, including hydrogels, are very important in terms of special applications, such as scaffolds and drug release. In this experiment, hydrogels were subjected to dynamic mechanical analysis using a compression mode. The obtained stress–strain (displacement) curves are illustrated in Figure 5. At the initial step, the compressive strength increases, depending on the PSS mole ratio; the higher the PSS mole fraction (f0.5-f0.8), the higher the initial compressive strength. It is observed that at the yield point, the further application of force causes a slight decrease in compressive strength. At this stage, it can be explained that the hydrogel cell walls start to displace by force due to the viscoelastic behavior of the PEC hydrogels. Then, the compressive strength gradually increases again, arising from the hydrogel cell wall being compacted by compressive force, as graphically drawn (Figure 6). The viscoelasticity can be further explained by loss modulus (E˝) and storage modulus (E’) plots (Figure 7). As can be seen, E′ at the initial step gradually increases (viscoelastic behavior) but then increases significantly with an increase in frequency, indicating an increase in the storage modulus due to the compact structure of the cell walls by force. On the other hand, the loss modulus (E˝) gradually increases, indicating the viscoelastic behavior of the PSS/PDADMAC hydrogel. The loss modulus/storage modulus ratios (E˝/E′) against frequencies (Hz) are plotted and shown in Figure 8; the lower the E˝/E′ value, the greater the compact structure of the hydrogel cell wall, as graphically illustrated in Figure 6. As can be seen in all cases, the E˝/E′ value gradually decreases with an increase in frequency, demonstrating that at higher frequencies, the PEC hydrogels exhibit a solid-like character due to the compact structure of the cell walls. The effect of frequency causes polyanions and polycations in the cell wall to vibrate, dissociating the electrostatic bonds (salt linkage). Under applied strain, polymer chains are forced to orientate along the force direction, resulting in a solid-like material. The viscoelasticity of the PEC hydrogels is interesting in terms of drug release under compressive conditions such as the stomach system.

### 3.6. Water Absorbency

The percent water uptake of freeze-dried PEC hydrogels was measured (compared to their dry weight). The results are compared and shown in Figure 9. From the resultant data, the swelling capability of the freeze-dried PEC hydrogels can be divided into two groups; the extremely high percent water uptake (f0.2-f0.4) and the low percent water uptake (f0.5-f0.8). The extremely high percent water uptake is found in the samples with high PDAMAC content due to its porous, thin cell wall containing macropores. As can be seen, the PEC hydrogels with a high content of PDADMAC are swollen and deformed due to their lack of dimensional stability, as illustrated in Section 3.1. The water absorption properties of the hydrogels are interesting in the fields of applications, including in biomedicals and water remediation.

### 3.7. Textile Dye Adsorption

The preliminary testing of the PSS/PDADMAC hydrogel on the removal of an anionic textile dye (C.I. Reactive Blue 49) is shown in Figure 10. The reactive dyes, which belong to the anionic dyes, are not only important dyes in the textile dyeing industry (which account for 80% of cellulosic dyeing), but are also the most environmentally unfriendly dyes, due to the removal difficulty of unreacted hydrolyzed color from wastewater [40]. Due to a high porosity and absorbency as well as an ionic–ionic interaction capability, the PSS/PDADMAC hydrogels exhibit the excellent removal of a representative anionic dye (Figure 10).

To elucidate the dye adsorption performance of the hydrogels, the batch adsorption test was carried out using a shaker. A series of 50 mL of initial dye concentrations (50 ppm, 100 ppm, and 150 ppm) were prepared. Then, 25 mg each of freeze-dried hydrogel samples (f0.2-f0.8) was added into an Erlenmeyer flask containing dye solution. The contact time vs. the adsorption capacity is plotted, as shown in Figure 11. The marked dye adsorption is found in the case of f0.2-f0.4 samples, where the hydrogels exhibit positive net charge, hence attracting to the anionic dye. In the case of f0.5 and f0.6 samples, which exhibit neutral charge and negative net charge, respectively, the anionic dye adsorption is much lower due to the absence of an attractive force (f0.5) and the effect of the repulsive force (f0.6). Moreover, a further increase in contact time leads to the dye desorption, due to the repulsive interaction among negatively charged dye molecules. For 100 ppm and 150 ppm, the dye adsorption trend on the f0.2-f0.4 samples is found in a similar manner to the 50 ppm dye solution, albeit exhibiting a higher adsorption capacity (Q_t_). In the case of the f0.4 sample (the highest dimensional stability compared to f0.2 and f0.3), the adsorption capacity at equilibrium (q_e_) is found to be 74 mg/g, 168 mg/g, and 245 mg/g for the initial dye concentrations of 50 ppm, 100 ppm, and 150 ppm, respectively. The adsorption performance of the PEC hydrogel toward the textile anionic dyes is attributed to the PEC positive net charge. Generally, the dyeing of cellulose is carried out in an alkaline condition. Therefore, the PSS/PDADMAC PEC hydrogel is an ideal candidate for textile wastewater decoloration due to its net charge that is independent of pH value, which was found in the polysaccharides-based polyelectrolyte complex hydrogel [6].

## 4. Conclusions

In this work, poly (sodium 4-styrenesulfonate) (PSS)/poly (diallyldimethylammonium chloride) (PDADMAC) complex hydrogels that had various PSS:PDADMAC mole fractions were prepared. SEM images showed that PSS/PDADMAC PEC hydrogels with high PSS mole fractions (f0.8-f0.7) comprised macropores (due to phase separation) and a thick and dense cell wall in the absence of micropores. In the case of a high PDADMAC mole ratio, the hydrogels exhibited macropores and a porous, thin cell wall (which was derived from water re-absorption into the thin cell wall, creating micropores). The hygroscopicity of the PE/PE complex cell wall was responsible for the water re-absorption ability. The BET surface area analysis revealed that the surface area decreased with a decrease in the PSS mole fraction, where the highest surface area of 3.5402 m^2^/g was found in the case of the f0.80 hydrogel and the lowest surface area of 0.4858 m^2^/g was found in the case of the f0.30 hydrogel. However, the BET surface area might not represent the true values due to the severe cleaning condition at a high temperature under vacuum, which caused sample damage, particularly to the delicate hydrogels (f0.4-f0.2). From the DMA results, the PSS/PDADMAC hydrogels exhibited viscoelastic behavior at a compressive force ranging from 0 to 0.1 N, causing the displacement of the hydrogel cell wall along the force direction. Finally, the PSS/PDADMAC hydrogels showed their high-water absorbency and affinity to textile anionic dyes. Based on the unique properties found, the PSS/PDADMAC hydrogel can be a potential candidate in the fields of tissue engineering, drug release, and wastewater treatment applications.

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
