# Peer review of "Preparation and Characterizations of PSS/PDADMAC Polyelectrolyte Complex Hydrogel"

_polymers, 2022, doi:10.3390/polym14091699_

Round 1

Reviewer 1 Report

This manuscript from Srikulkit et all introduces the synthesis of PSS/PDADMAC hydrogels from polyelectrolyte complexation. While the synthesis approach is original and PEC-based materials are of high interest to the audience, there are a number of concerns (listed below) that cannot be adressed in a reasonnable schedule. I therefore recommand to reject this paper and encourage its resubmission. 

The first synthesis step is based on the electrostatic complexation of polyelectrolytes at different molar ratios  (solution pH and ionic strength unspecified), yielding aggregates to be later redissolved in a high ionic strength solution (NaCl 4M). Note that this step is decribed in the introduction but not in the materials and method section so I am unsure of what has been done:

  • If the polyelectrolytes were indeed mixed before redissolving in 4M NaCl, complexation surely happened. In that case, the community is well aware that the molar ratio of polyelectrolytes in the PEC does not always reflect the molar ratio  used during synthesis. Authors should characterize the exact composition of their complexes, as well as that of their hydrogels (later in the synthesis process). Elemental analysis of Sulfur and Nitrogen for all synthesis batches would be a good way to proceed. 
  • If the polyelectrolytes were mixed directly in a 4M NaCl solution (sometimes 3M is also reported whithout explaining why) , one can doubt that there is any complexation/aggregation occuring between polymers. Did authors perform turbimetry or light diffusion measurements supporting that complexation occured ?

Additional questions are related to the materials'analysis and function:

  • Freeze drying of hydrogels was performed before their SEM analysis: isn't this approach going to influence (if not control !) the porosity of the gel ?
  • The pore size distributions on micrographes of Figure3 should be analyzed (for instance by using ImageJ)
  • The porous morphology of PEC-based materials is dramatically sensitive to their environment's ionic strenght. Therefore, comparing the pore structure of hydrogels (figure 3) whithout knowing their NaCl content is not relevant. A detailled analysis of the NaCl removal process by dialysis shoud be demonstrated in this paper.
  • It is clear that the pores in the gel are of macroporous size. BET analysis on such materials is largely irrelevant (especially to calculate a pore diameter !)
  • It is not clear whether the "polyelectrolyte ratios" (cf my remark above) in the gel has any effect on the mechanical properties presented on figure 4 and 6. More generally the discussion of these 2 figures is poorly developped.
  • The drug relese model proposed on figure 7 is not supported by anay experimental evidence
  • The textile dye adsorpion is not discussed in connection with any litterature nor benchmark

Finally, althought I don't feel qualified to juge the english level of this paper, I recommend authors to perform an extensive revision on this matter

Author Response

This manuscript from Srikulkit et all introduces the synthesis of PSS/PDADMAC hydrogels from polyelectrolyte complexation. While the synthesis approach is original and PEC-based materials are of high interest to the audience, there are a number of concerns (listed below) that cannot be adressed in a reasonnable schedule. I therefore recommand to reject this paper and encourage its resubmission. 

The first synthesis step is based on the electrostatic complexation of polyelectrolytes at different molar ratios  (solution pH and ionic strength unspecified), yielding aggregates to be later redissolved in a high ionic strength solution (NaCl 4M). Note that this step is decribed in the introduction but not in the materials and method section so I am unsure of what has been done:

Answer to comment:

Thank you for comments. We have re-written the preparation of PEC hydrogel (section 2.2) more clearly as highlighted. In summary, PEC hydrogel was synthesized in one bath (aggregate formation in the presence of salt, then re-solubilization by heating/stirring). Finally, the hydrogel was obtained desalting method using dialysis membrane. pH is not a factor in case of PSS/PDADMAC hydrogels since PSS/PDADMAC are present in ionized forms over a wide range of pH values (pH 1 and above). Therefore, solution pH does not interfere with aggregation or solubilization. Effects of salt concentration on PEC solubility test is added into the manuscript.

  • If the polyelectrolytes were indeed mixed before redissolving in 4M NaCl, complexation surely happened. In that case, the community is well aware that the molar ratio of polyelectrolytes in the PEC does not always reflect the molar ratio  used during synthesis. Authors should characterize the exact composition of their complexes, as well as that of their hydrogels (later in the synthesis process). Elemental analysis of Sulfur and Nitrogen for all synthesis batches would be a good way to proceed. 

Answer to comments.

Thank you for comments. We are bear in mind that calculation of mole ratio based on polymer molecular weight will get a false result since polymer is not a single molecule. For example, PDADMAC has a wide range of Mw from 20000-350000 g/mole. We are aware of the right method of mole ratio calculation. In this experiment, we calculated mole ratio from mole ratio of repeating unit (monomers). We did put asterisk sign (*) to indicate clearly. 

  • If the polyelectrolytes were mixed directly in a 4M NaCl solution (sometimes 3M is also reported whithout explaining why) , one can doubt that there is any complexation/aggregation occuring between polymers. Did authors perform turbimetry or light diffusion measurements supporting that complexation occured ?

Answer to comments.

Thank you for comments. Different concentration is obtained from re-solubility test as mentioned in experimental section. From the draft manuscript, we do apologize for confusion. Actually, aggregation/re-solubilization in the presence of salt was carried out in one-bath experiment.  We rewrote this experiment part as highlighted.

Additional questions are related to the materials'analysis and function:

  • Freeze drying of hydrogels was performed before their SEM analysis: isn't this approach going to influence (if not control !) the porosity of the gel ?

Answer to comments.

Freeze-drying is a must sample preparation to investigate porosity of hydrogels. Freeze-drying is a technique for sublimation (not evaporation) of iced water crystal. When iced water crystal is sublimated, porous structure is observed while the cell wall skeleton is maintained. Note that wet hydrogel is allowed for SEM instrument since the evaporation of water not only damage the instrument but also condense onto glass window. SEM image cannot be taken.

  • The pore size distributions on micrographes of Figure3 should be analyzed (for instance by using ImageJ)

Answer for comment.

Thank you for comment. Image J was used to measured macropore sizes which were summarized in Table 2.

  • The porous morphology of PEC-based materials is dramatically sensitive to their environment's ionic strenght. Therefore, comparing the pore structure of hydrogels (figure 3) whithout knowing their NaCl content is not relevant. A detailled analysis of the NaCl removal process by dialysis shoud be demonstrated in this paper.

Answer to comment.

We provide a detailed explanation in the revised manuscript as highlighted. In summary, desalting using dialysis method is reversible process when it reaches equilibrium time. So that the wording “ NaCl removal” is not fit to dialysis. In this type of hydrogel formation, we need both diffusion into dialysis bath and diffusion from dialysis bath back into membrane tube to control phase separation which creates hydrogel cell wall. Dialysis water bath was changed every interval time of 2 h for 1 day to obtain viscoelastic hydrogel. I am not sure I satisfy reviewer question or not.

  • It is clear that the pores in the gel are of macroporous size. BET analysis on such materials is largely irrelevant (especially to calculate a pore diameter !)

Answer for comment.

Thank you for your comment. We agree with this comment. BET surface area analysis is too severe for our delicate samples. It is likely that cell wall was collapsed during surface cleaning step. Therefore, BET surface area determination does not represent the intrinsic surface area as highlighted in the revised manuscript. 

  • It is not clear whether the "polyelectrolyte ratios" (cf my remark above) in the gel has any effect on the mechanical properties presented on figure 4 and 6. More generally the discussion of these 2 figures is poorly developped.

Answer to comment.

We have intensively revised this discussion section according to your comment.

  • The drug relese model proposed on figure 7 is not supported by anay experimental evidence

Answer to comment.

This figure was removed.

  • The textile dye adsorpion is not discussed in connection with any litterature nor benchmark

Answer to comment.

We cite our own work ref. 40 concerning cellulose dyeing and refer to ref. 6 for benchmarking. However, we avoid saying that our material is better.

Finally, althought I don't feel qualified to juge the english level of this paper, I recommend authors to perform an extensive revision on this matter

Answer to comment.

We did carefully re-check writing.

Author Response

The authors prepared PSS/PDADMAC PEC hydrogel by dialyzing salt-added PEC solution. Porous morphology, viscoelasticity, water absorbency, textile dye adsorption of hydrogel were studied. The results are interesting, and it would be useful information for the future applications. However, following issues should be addressed.

  1. In Table 1, why different NaCl concentrations were used instead of fixed NaCl concentration? 4N for f0.5-f0.8 and 3N for f0.2-0.4.Do you expect the same results if you used 4N or 3N for all the samples?

Answers to comment.

Thank you for comment. We provide additional data concerning this matter in section 2.2. In summary, due to electrostatic strength is varied different amount of salt concentration is required to obtain complete re-solubility of PEC aggregate.

  1. In line 125-128, dimensional stability of f0.6-0.8 hydrogel was attributed to repulsive interaction among PSS, and deformed structure of f0.2-0.4 was attributed to diminish of repulsive interaction of PSS. However, isn’t there repulsive interaction among PDADMAC in f0.2-f0.4?

Answers to comment.

Thank you for comment. Dimensional stability explained by repulsive interaction is ambiguous. Therefore, we rewrite the discussion of dimensional stability with relation to hydrogel cell wall and cell wall porosity. Please be referred to changes as highlighted in the revised manuscript.

  1. In line 129-130, the authors also mentioned hygroscopicity of PDADMACto explain deformed structure. Do you have any reference or experimental data that compares the hygroscopicity of PDADMAC and PSS?

Answers to comment.

Thank you for comment. We proposed the mechanism of hydrogel formation in section 3.1. Hydrogel formation involves phase separation to create cell wall and macropore at the first stage. Then, water re-absorption into cell wall is dependent on hygroscopicity (PDADMAC content) of cell wall, producing a hierarchical porous cell wall.

  1. Hard to understand the compact structure of Figure 5. Compact structure looks like the structure before the sample was compressed.

Answers to comment.

Thank you for comment. Model of compact structure of cell wall is re-drawn as shown in Figure 6 (revised version).

  1. In Figure 6, why f0.5 sample has high E”/E’(more damping) than f0.3 sample at a given frequency? In my understanding f0.3 should be more viscous(higher E”/E’). Can you also plot both E’ vs frequency and E” vs frequency?

Answers to comment.

Thank you for comment. E”/E’ plots are given. F0.5 is solid-like hydrogel. F0.3 is dimensionally instable hydrogel. In my opinion, those two samples cannot be comparable.

  1. Please provide f0.3 data in Figure 19.

Answers to comment.

Thank you for comment. Data are provided in Figure 11 (revised).

Reviewer 3 Report

​There are some questions that need to be addressed to make this work publishable.

  • 56 Authors mention pH dependency of PEC. Is this system pH dependent? Was the pH controlled in all experiments?  
  • 82 Why do we need 70 C for PEC formation?
  • 111 What is the detection wavelength? How was Qt  (mg/g) calculated from the absorbance? Calibration curve?
  • 191 Could authors provide more information regarding the controlled release? Is it in water? What kind of compressive conditions? What type of drugs?
  • 198 During modulus calculations was the difference in porosity taken into account?
  • Authors mention (207 for example) hygroscopicity of PDADMAc. How was it compared to PSS hidrofilicity? Is there any literature data?  
  • 212 What is the origin of such a significant difference in behavior for f0.5, f0.6 and  f0.4 hydrogels? The difference in ratio is not that high, bit properties are drastically changed.  
  • Have you tested f0.3? It should have even higher dye removal properties
  • The main question: Why don't we see the reverse of all properties when we switch charge ratio? Why does the excess of negative net charge lead to mechanically more stable samples whereas net positive - not?

Author Response

Comments and Suggestions for Authors

​There are some questions that need to be addressed to make this work publishable.

  • 56 Authors mention pH dependency of PEC. Is this system pH dependent? Was the pH controlled in all experiments?  

Answer to comment.

Thank you for comment. Actually, PSS/PDADMAC PEC hydrogel formation is independent to pH value. So that we omit “pH value” from line 56.

  • 82 Why do we need 70 C for PEC formation?

Answer to comment.

In re-solubilization step, heating and stirring were required. 70 C is a minimum temperature found to obtain homogenous PEC solution.  We highlight explanation in the revised manuscript.

  • 111 What is the detection wavelength? How was Qt  (mg/g) calculated from the absorbance? Calibration curve?

Answer to comment.

We include the detail as suggested in the revised manuscript. Wave length is 590 nm and Qt is measured from absorbance value at  maximum wave length.

  • 191 Could authors provide more information regarding the controlled release? Is it in water? What kind of compressive conditions? What type of drugs?

Answer to comment.

Since in this manuscript drug release model is proposed without experimental data support. So that we decide to remove Fig. 7 from the revised manuscript.

  • 198 During modulus calculations was the difference in porosity taken into account?

Answer to comment.

Yes, as highlighted in the revised manuscript.

  • Authors mention (207 for example) hygroscopicity of PDADMAc. How was it compared to PSS hidrofilicity? Is there any literature data?

Answer to comment.

 We rewrite the discussion concerning this matter in section 3.3. In summary, cell wall of hydrogels with higher mole fraction of PDADMAC is thin and porous due to re-absorption of water into cell wall arising from PDADMAC hygroscopicity.

  • 212 What is the origin of such a significant difference in behavior for f0.5, f0.6 and  f0.4 hydrogels? The difference in ratio is not that high, bit properties are drastically changed.

Answer to comment.

As explained in the revised manuscript, the formation of hydrogel cell wall with varying pore sizes is governed by net charge balance (f0.5), negative net charge (f0.6) and positive net charge (f0.4).

  • Have you tested f0.3? It should have even higher dye removal properties

Answer to comment.

Yes, f0.3 exhibits higher dye removal. The result is added into the revised manuscript.

  • The main question: Why don't we see the reverse of all properties when we switch charge ratio? Why does the excess of negative net charge lead to mechanically more stable samples whereas net positive - not?

Answer to comment.

Yes, our results are found in the same trend to ref. 38.  We also provide explanation on this matter as follows:

  1. We also found that high PSS content led to thick and dense cell wall. However, the cell wall exhibited no porosity as found in case of net charge balance and positive net charge. An explanation is that water re-absorption into thick and dense cell wall is limited as explained by hydrophobic aromatic pendant groups of PSS ( pi-pi interaction of aromatic pendant group (ref 38). Therefore, cell wall synthesized by these recipes is non-porous cell wall.
  2. On the other hand, things are opposite in case of net charge balance point and positive net charge where porous cell wall is observed. More explanation is referred to revised manuscript.

Round 2

Reviewer 1 Report

The changes made by authors do not address the 2 main issues identified in the previous version of this manuscript that is: 

  • The polyelectrolyte ratio in the final material is unknown as only the mixing conditions are controlled. Complexes are not necessarily going to reflect the mixing ratio . And subsequent treatments (ionic strength changes and dialysis) are capable of changing further this ratio.
  • The salt content of the final material (after dialysis) is unknown as the dialysis process has not been monitored.
  •  
  • These 2 parameters (polyelectrolyte ratio and salt content) were previously described as critical for both the porosity and the mechanical properties of the polyelectrolyte complex materials. As the present work has no control over such prominent parameters, I cannot recommend this article for publication. 

Reviewer 2 Report

The manuscript has been well revised.